# Anti-CD20 Agents for Multiple Sclerosis: Spotlight on Ocrelizumab and Ofatumumab

**DOI:** 10.3390/brainsci10100758

**Published:** 2020-10-20

**Authors:** Despoina Florou, Maria Katsara, Jack Feehan, Efthimios Dardiotis, Vasso Apostolopoulos

**Affiliations:** 1Neurology Department, University Hospital of Larissa, University of Thessaly, 41110 Larissa, Greece; despoina_flor@yahoo.com; 2Therapeutic Area Head Neuroscience & Ophthalmology, Novartis (Hellas) S.A.C.I., Medical Department, 14451 Athens, Greece; maria.katsara@novartis.com; 3Department of Medicine, Western Health, The University of Melbourne, Melbourne 3010, Australia; jfeehan@student.unimelb.edu.au; 4Institute for Health and Sport, Victoria University, Melbourne 8001, Australia

**Keywords:** monoclonal antibodies, multiple sclerosis, B-cell therapies, safety

## Abstract

Until recently, in the pathogenesis of Multiple Sclerosis (MS), the contribution of B cells has been largely underestimated, and the disease was considered a T-cell-mediated disorder. However, newer evidence shows that B cells play a crucial role in the pathogenesis of MS via antigen-driven autoantibody responses and through the cross regulation of T-helper cells. As B cells express the surface molecule CD20 at all points of differentiation, it provides a specific target for monoclonal antibodies, and the development and clinical testing of anti-CD20 antibody treatments for MS have been successful. After some observations, some small clinical trials found positive effects for the first anti-CD20 therapeutic rituximab in MS; newer agents have been specifically evaluated, resulting in the development of ocrelizumab and ofatumumab. Ocrelizumab, a humanized anti-CD20 monoclonal antibody, was approved in March 2017 by the Food and Drug Administration (FDA) and is also the first proven therapy to reduce disability progression in primary progressive MS. This is particularly significant considering that disease-modifying treatment options are few for both primary and secondary progressive MS. Ofatumumab, a fully human anti-CD20 monoclonal antibody, that binds a distinct epitope, has been further investigated in phase 3 trials for relapsing forms of MS. In this review, we discuss in detail these two anti-CD20 agents and their advent for treatment of MS.

## 1. Introduction

Multiple sclerosis (MS) is a chronic autoimmune demyelinating disorder of the central nervous system (CNS) characterized by the accumulation of inflammatory cells and cytokines at the sites of demyelination and plaque formation [1]. MS is known for its chronic degenerative effect on the neurological system leading to diverse progression patterns, giving rise to a number of phenotypes. MS is broadly categorized into three major subtypes: relapsing–remitting MS (RRMS), progressive MS, and clinically isolated syndromes (CIS) [2]. RRMS is characterized by episodes of disease activity, with new or expanding CNS lesions, while progressive disease is characterized by a gradual worsening of clinical symptoms over time which may or may not be accompanied by inflammatory activity. The majority of those diagnosed with MS have RRMS (85%) which, over time in some patients, progresses to secondary progressive MS (SPMS), in which worsening disease leads to progression of clinical symptoms. A section of patients (10–15%) are diagnosed with primary progressive MS (PPMS); this subgroup is treated as a distinct type of MS, characterized by ongoing clinical deterioration [2]. The symptoms of MS and the progression of the disease vary, with some individuals having very few symptoms whilst others require a range of assistive therapies and mobility aids for function such as walkers or wheelchairs [3]. Important clinical measures for MS include the Expanded Disability Status Score (EDSS), which is a measure of a broad range of impairments including tremor, weakness, incontinence, and speech difficulties, among others; the relapse rate, which measures the speed of new neurological symptom onset over time; and magnetic resonance imaging (MRI) findings, such as gadolinium-enhancing lesions which indicate inflammation of a plaque, or T2 lesions which indicate broader MS plaque volume. These measures are critical to clinical decision making, particularly in the realm of drug prescription and treatment. The progression of disability, inflammatory disease activity, clinical relapse, and radiographical changes in the CNS are all indicators for particular therapeutic interventions, and should guide decision making. The ultimate goal of modern MS therapies is to achieve no evidence of disease activity (NEDA) in which therapy has halted relapses and disability progression, as well as new and active MRI lesion development. There are a number of drugs available for MS patients; however, while they decrease the relapse rate of RRMS, safety concerns, individual immunological changes, and issues with compliance make their long-term use challenging. Effective treatments for the progressive forms of MS are more limited, with only a small number of therapeutic agents available with beneficial effects. Hence, there is a requirement for new improved treatment modalities to effectively treat MS [4,5,6].

## 2. T- and B-Cell Hypothesis

MS is characterized as a complex inflammatory disease, where T helper (Th) 1, Th2, Th9, Th17, Th22 cells, CD4+, CD8+ T cells, B cells, and M1 macrophages infiltrate the sites of neuronal, blood–brain barrier, myelin damage, and glial activation [4]. This immune response results in cytokine secretion (Interleukin [IL] -2, IL-6, IL-9, IL-12, IL-17, IL-22, IL-23, IL-27, interferon [IFN]-gamma, tumor necrosis factor [TNF]-alpha) and activation of an inflammatory storm leading to axonal damage, gliosis, disruption of the blood–brain barrier, and demyelination within the brain and spinal cord [7,8,9]. Thus, it is appropriate to target specific immune cells and/or their constituents rather than global immunosuppression to reduce side effects and modulate inflammation. Historically, MS was seen as a T-cell pathology; however, it is now known that B cells play a large role in the pathogenesis of the disease. Dysfunctional regulation of B cells and Th cells leads to ongoing autoimmune activation against the myelin sheath. Lymphocytes can cross the blood–brain barrier, through a complex process of extravasation, and enter the brain parenchyma. Initially it is thought that an autoimmune trigger causes Th-cell trafficking to the CNS, which in turn activates and attracts B cells to the affected area. These B cells acquire self-antigens, differentiate to antibody-producing plasma cells, and begin the production of auto-antibodies against neural tissues. B cells also co-regulate the Th cells, leading to further immune cell trafficking to the region of dysfunction. A majority of MS patients are known to have accumulation of both immune cells, auto-antibodies, cytokines, and inflammatory mediators in the cerebrospinal fluid, particularly during times of relapse or increased disease activity, clear evidence of an accelerated inflammatory immune response underpinning its pathogenesis. The immune cells which have migrated to the CNS aggregate in an organized tertiary ectopic follicle, which are known to remain active; however, their contribution to the ongoing disease is not yet fully understood [10]. While the role of B cells in the pathogenesis of MS is clear, there are still many unanswered questions. The factors which are involved in their trafficking to the CNS, the exact mechanisms by which they cross the blood–brain barrier, as well as their behavior after the cessation of the inflammatory period are yet to be fully understood. Despite these unknowns, the success of B-cell depleting therapies [11] has reinforced the modern understanding of the pathophysiological perception of MS, as a disease of both T and B cells [12,13].

B cells are classified as regulatory B cells (which secrete IL-10, IL-35) and, importantly in MS, as pro-inflammatory B cells (which secrete IL-6, IL-12, IL-15, TNF-alpha, granulocyte macrophage colony stimulating factor [GM-CSF]) that regulate T-cell polarization towards pro-inflammatory immune responses (Figure 1) [12,14]. In addition, B cells, alongside other antigen-presenting cells, play a highly effective role in antigen processing and presentation as they express the major histocompatibility complex and T-cell activation surface markers CD80, CD86 leading to further activation of T cells in MS (Figure 1) [15,16]. Furthermore, in MS, B cells are clonally expanded and secrete autoantibodies (IgG) with oligoclonal bands noted in the cerebrospinal fluid and brain parenchyma in most (>90%) patients [17]. Antigen-specific memory B cells are also present in the central nervous system of patients with MS, and are detected in MS lesions [18,19]. Clearly, B cells are present within the central nervous system in individuals living with MS, suggesting that they play a major part in the pathophysiology and course of disease. Depletion of B cells has been shown to be a viable approach as a therapeutic target against MS.

B cells can be identified by a range of cellular markers, which also allow differentiation of important subsets. The B-cell lineage expresses the antigen CD20 at all points, meaning that these markers are able to identify cells at all states of maturity and differentiation. The majority of B cells also express CD19, with the exception of the antibody-secreting plasma cell, making it a less attractive therapeutic target. They also express markers shared by a range of other immune cells, such as CD32 and CD35. Other markers can be used alongside these to identify other important B-cell subsets, such as memory B cells (CD27+) and plasma cells (MHCII+). In recent years, CD19 and CD20 have provided to be viable targets for selective targeting for monoclonal antibodies [16]. CD19 is expressed on all B-cell lineage cells except for plasma cells and its main function is to decrease signaling via the B-cell immunoglobulin receptor. As CD19 is highly expressed on B cells, it has been used as a target for immunotherapy against hematological malignancies [20]. CD20 is a transmembrane protein spanning 4 regions, is highly expressed on the surface of B cells, and plays a role in their differentiation into plasma cells and activation of antigen independent T-cell responses. CD20 is expressed by B-cell lymphomas and leukemias and some myelomas, thymomas and Hodgkin’s disease [21,22]. While classically CD20 was thought to be exclusively a B-cell marker, it has also been identified on a small subset of T cells; however, it is unclear whether their depletion is responsible for some of the effects of anti-CD20 therapies [23]. Using antibody therapies to deplete B cells allows the pathogenic cycle of inflammation and immune-mediated CNS injury to be broken and allow for longer periods between relapses. This comes alongside a host of clinical benefits, with a decreased and slowed progression of disability. There is growing research interest surrounding the use of anti-CD20 antibodies as a treatment for all forms of MS [24,25]. Targeting CD20 using monoclonal antibodies has shown to be revolutionary against B-cell leukemias and lymphomas and more recently against autoimmune diseases where B cells play a fundamental role, such as systemic lupus erythematosus and MS [26,27].

## 3. Anti-CD20 Agents

A number of anti-CD20 monoclonal antibodies have been approved and used in human clinical trials, primarily, obinutuzumab, tositumomab, ublituximab, ocrelizumab, ofatumumab and rituximab. Briefly, obinutuzumab (also known as afutuzumab, GA101; by GlycArt Biotechnology AG and Roche) is a humanized monoclonal antibody (from mouse) used intravenously against chronic lymphocytic leukemia, non-Hodgkin’s lymphoma, diffuse large B-cell lymphoma, and was approved by the Food and Drug Administration (FDA) and the European Medicines Agency (EMA) in 2013 under the trade names Gazyva and Gazyvaro, respectively [28]. Tositumomab (also known as Bexxar; by GlaxoSmithKline) is an ^131^-Iodine-labeled murine monoclonal antibody that was approved for use in non-Hodgkin’s lymphoma in 2003 [29]. The radiolabeled antibody was used to increase the radiation delivered to the tumor with less overall body radiation exposure; however, in 2014, Bexxar was discontinued due to lack of demand as newer, improved anti-CD20 antibodies were available. Ublituximab (TG-1101; by TG Therapeutics) is a chimeric glycol-mouse/human anti-CD20 monoclonal antibody currently in phase II human clinical trial for chronic lymphocytic leukemia. In addition, a phase III randomized, multicenter, double-blind clinical trial is being compared against teriflunomide, and is currently recruiting patients with RRMS with an estimated completion date of 30 September 2021 (registration no. NCT03277261). In MS more broadly, there are three major anti-CD20 therapies available, Rituximab, Ocrelizumab, and Ofatumumab [30].

The first anti-CD20 therapy to be used in MS is the chimeric antibody Rituximab. Rituximab (also known as Rituxan or MabThera; by Biogen and Genentech, USA and Hoffmann-La Roche, Canada) was approved in 1997 for some cancers and for autoimmune disorders and is on the WHO’s list of essential medicines [31]. While rituximab is not FDA- or EMA-approved for use in MS, it is commonly prescribed off-label, particularly in resistant cases. Rituximab biosimilar compounds are currently under investigation, with a number currently under investigation [32]. Despite its common use in MS, there are relatively few high-quality studies surrounding its efficacy. In the first phase I clinical trial of rituximab, named HERMES, 69 patients with RRMS were given 1000 mg of rituximab, and compared against placebo-treated controls [33]. The patients receiving rituximab had reduced total, and new gadolinium-enhancing lesions at 12, 16, 20, and 24 weeks compared to controls (*p* = < 0.001). The patients receiving the treatment also had fewer relapses than controls at both 24 and 48 weeks (14.5% vs. 34.3%, *p* = 0.02) and a decrease in the annualized relapse rate (ARR) at 24 weeks (0.37 vs. 0.84, *p* = 0.04), but not at 48 weeks. The phase I HERMES trial was followed up by a phase 2/3 trial of 439 patients with PPMS named OLYMPUS which evaluated the effect of Rituximab on disease progression [34]. Note that 292 patients received 1000 mg of rituximab at weeks 0, 2, 24, 26, 48, 50, 72, and 74, and were compared against placebo-treated controls. The trial demonstrated mixed results, with no difference in time to disease progression, brain volume loss, or a number of functional criteria of the multiple sclerosis functional composite. There was however a decrease in T2-weighted lesion volume on MRI, and an improvement in 25-foot walk test. While level 1 evidence is lacking, there are a small number of observational and cohort studies which support the use of rituximab in MS. These studies have shown that the medication appears to improve MRI and clinical findings in RRMS, PPMS, and SPMS, with reduced ARR, T2, and gadolinium-enhancing lesions [35,36,37,38,39]. The clinical studies on rituximab are summarized in Table 1. While on balance the evidence suggests that rituximab is likely to have beneficial effects, particularly in RRMS, in light of the newer agents ocrelizumab and ofatumumab becoming more broadly available and having stronger therapeutic efficacy, interest in its use is declining. There are however currently ongoing clinical trials, which may provide new evidence of positive effect.

## 4. Ocrelizumab

Ocrelizumab (also known as Ocrevus^®^; marketed by Genentech, US) is a humanized anti-CD20 monoclonal antibody (from mouse) which binds to an overlapping epitope to that of rituximab and is used intravenously. Ocrelizumab specifically depletes B cells and was approved by the FDA in March 2017 for the treatment of MS, in particular those with active PPMS [40,41]. The EMA approval followed 9 months after the USA approval, and it introduced a new era of B-cell-targeted treatment of MS. The acceptance of anti-CD20 agents is reflected in the market share of Ocrevus^®^, which accounted for an estimated 13.8% of MS medication sales a year after release [42].

### 4.1. Dosage and Administration

The starting concentration to be used is generally 300 mg in 250 mL 0.9% NaCl (i.e., 1.2 mg/mL), followed by a second injection after 2 weeks, and thereafter every 6 months of 600 mg in 500 mL 0.9% NaCl, intravenously. Prior to ocrelizumab injection, patients are required to be premedicated at least 30–60 min prior with 100 mg methylprednisolone (or an equivalent) and an antihistamine, in order to avoid infusion-related reactions. Patients are required to be observed for at least 60 min following ocrelizumab infusion [43].

### 4.2. Pharmacology and Pharmacokinetics

Ocrelizumab is a humanized anti-CD20 IgG1 mAb which binds to the extracellular loop of CD20, causing antibody-dependent cell lysis in circulating B cells. The mechanisms of apoptotic B-cell depletion with ocrelizumab use are antibody-dependent cell-mediated phagocytosis, antibody-dependent cellular cytotoxicity (ADCC), as well as complement-dependent cytotoxicity (CDC) [43,44]. Interestingly, recent studies have shown that ocrelizumab may also target CD20+ T cells, which are shown to be present in MS patients, suggesting an alternative contributing mechanism [23]. Ocrelizumab shows effectively linear, dose-proportional pharmacokinetics for doses of 400 mg to 2000 mg. Ocrelizumab, as with other antibodies, undergoes nonspecific catabolism and degrades into smaller peptides and amino acids when metabolized, with a half-life of 26 days [41].

### 4.3. Clinical Trials

The following studies have examined the efficacy and safety of ocrelizumab in RRMS patients, as shown in Table 2. In a phase 2, 48-week, double-blind, randomized parallel, placebo-controlled trial, 220 patients with RRMS received two doses of either an intravenous placebo, a low dose (600 mg), or a high dose (2000 mg) of ocrelizumab on days 1 and 15, alongside a control group receiving a weekly 30 μg dose of interferon beta-1a (IFN-beta-1a) [45]. After 24 weeks, there was a significant reduction in the number of gadolinium-enhancing lesions, with decreases of 89% in the low-dose group and 96% in the 2000 mg group, with no difference in serum IgG. Infusion-related adverse effects, most often infections and headache, were reported as more serious in the 2000 mg group compared to patients receiving the lower dose, but incidence of these side effects was equivalent in both groups (35% in the 600 mg vs. 44% in the 2000 mg group in cycle 1, and 16% vs. 17% in cycle 2). Only 9% of RRMS patients tested with IFN-beta-1a experienced infusion-associated adverse effects, and there was one death reported on trial in the high-dose ocrelizumab group after complications following a bee sting.

Two equivalent phase 3, randomized and double-blind clinical trials named OPERA 1 and 2 were conducted to test the efficacy of ocrelizumab versus IFN-beta-1a in RRMS patients [46]. The two studies had the same duration (96 weeks) and randomized 821 and 835 RRMS patients, aged 18 to 55 years. The participants received either 600 mg of intravenous ocrelizumab every 24 weeks or 44 μg of subcutaneous IFN-beta-1a three times weekly. The group given the ocrelizumab received their first dose as two infusions of 300 mg each, given two weeks apart, with the remaining doses at the 24 week interval. The trials included patients with an Expanded Disability Status Score (EDSS) between 0 and 5.5 and who had suffered more than two clinical relapses in the two years prior to the study, or at least one in the past year. Both studies showed a significant decrease in the annualized relapse rate (ARR) for patients treated with ocrelizumab at 96 weeks than those treated with IFN-beta-1a (0.16 versus 0.29, *p* = < 0.001). The percentage of patients with progression in their disability was lower at both 12 (9.1 vs. 13.6% *p* = < 0.001) and 24 weeks (6.9 vs. 10.5% *p* = 0.003). There was also a 94% decrease in gadolinium-enhancing lesions with ocrelizumab compared with placebo in both trials. The mean new or newly expanding lesions on T2-weighted MRI imaging was significantly lower in the group given ocrelizumab versus the interferon beta-1a group in both trials.

In a 120 week, double-blind and placebo-controlled phase 3 trial named ORATORIO [47], a total of 732 patients with PPMS were randomized to receive 600 mg of ocrelizumab intravenously or placebo every 24 weeks at a ratio of 2:1. Participants included were aged 18–55 years with a diagnosis of PPMS through the McDonald criteria [48], characterized by oligoclonal bands suggestive of elevated IgG in the CSF and had an EDSS score of 3.0–6.5 at screening [47]. The patients receiving ocrelizumab had lower disability progression at week 12 (32.9% vs. 39.3%, *p* = 0.03) and 24 (29.6% vs. 35.7%, *p* = 0.04) compared to placebo. The ocrelizumab group also showed significant improvements in brain volume loss when compared to placebo (0.90% vs. 1.09%, *p* = 0.02).

Following the approval of ocrelizumab by the FDA in March 2017, Genentech has been required to undertake a phase IV human clinical trial in young adults to determine whether any side effects such as increased risk of cancer or effects in pregnant women and their newborns result [49]. However, in mouse studies it was shown that the drug was present in breast milk, and hence, it is unclear what effects this may have in newborns, so it is recommended not to be used while breast feeding.

## 5. Ofatumumab

Ofatumumab is an anti-CD20, human monoclonal IgG1 antibody binding strongly to a distinct membrane epitope to rituximab and ocrelizumab [41,50]. Ofatumumab is the first type 1 immunoglobulin G1 kappa (IgG1κ) monoclonal antibody that is fully human, and is currently licensed for the treatment of Chronic Lymphocytic Leukemia under the brand name Arzerra [50], but it has also recently been evaluated for use in RRMS. Novartis acquired the rights from GlaxoSmithKline (GSK) in 2015, to allow for the development of ofatumumab for cancer and autoimmune diseases. A notable strength of ofatumumab is that it can be administered subcutaneously by patients or caregivers with an auto-injector pen, which is administered at four-week intervals. In the face of a chronic disease requiring regular treatment administration, this may provide better access to therapy, as opposed to typical antibody therapies which necessitate a day to be set aside in a clinic for the infusion. In August 2020, the FDA approved ofatumumab as a therapy for all forms of relapsing MS, including CIS, secondary progressive MS and RRMS, in the form of an auto-injector pen. It is expected to receive approval from the EMA in 2021.

### 5.1. Pharmacology

Ofatumumab binds to two novel epitopes of the CD20 protein: the small and large extracellular loops. Once bound by the antibody, the CD20 molecule is not internalized or shed from the cell surface [24]. As with other anti-CD20 antibodies, after binding target epitopes, ofatumumab causes antibody-dependent cell lysis to the circulating B cells [51]. The mechanisms of cell lysis are similar to ocrelizumab, although ofatumumab causes more CDC than ADCC [52].

### 5.2. Clinical Trials

Ofatumumab has demonstrated strong efficacy in rheumatoid arthritis [53] and hematological malignancies [54]; however, some studies have also evaluated its role in treating MS (Table 3).

In a small phase 2, double-blind, randomized, placebo-controlled study, 38 patients with RRMS received either intravenous ofatumumab or a placebo at two-week intervals [55]. A group of 26 patients received fortnightly 100, 300, and 700 mg doses of ofatumumab, while the remaining 12 were placebo-treated. After 24 weeks in two groups, the two treatment arms were swapped, with the placebo participants receiving ofatumumab. A 99% reduction of new T1 and T2 gadolinium-enhancing brain lesion activity after 24 weeks of ofatumumab treatment was reported. Moreover, no reduction in overall serum IgG was observed, and ofatumumab was well tolerated by patients. A second phase 2, 48 week, double-blind, placebo-controlled, randomized and manufacturer-sponsored study named MIRROR was conducted to examine subcutaneous ofatumumab versus placebo in 231 patients with RRMS [56]. The study randomized participants to receive placebo (*n* = 67), 12 weekly doses of either 3 mg (*n* = 33), 30 mg (*n* = 32), or 60 mg (*n* = 33) of ofatumumab, or finally, 60 mg ofatumumab every 4 weeks (*n* = 63). A dose of 3 mg ofatumumab was given to the placebo group at week 12 with a second at 24 weeks during a follow-up phase. The study confirmed that all doses of ofatumumab significantly decreased new MRI lesions over 12 weeks. Infection-associated AEs were reported in both groups, but were non-significantly different (25% placebo, vs. 27% ofatumumab at 12 weeks, 29% placebo and 21% ofatumumab at 24 weeks). An extension of this trial also showed ongoing suppression of new lesions after 48 weeks with dose-responsive B-cell effects.

Two phase 3 randomized, double-blind, active-controlled studies (ASCLEPIOS 1 and ASCLEPIOS 2) have investigated the efficacy of ofatumumab versus teriflunomide in RRMS. The trials reached their end points, with positive effects reported compared to teriflunomide in relapse rate, confirmed disease improvement, and number of T1 and T2 gadolinium-enhancing lesions [57]. The safety profile of ofatumumab was generally good, with similar incidence of AEs in patients receiving it compared to those who received teriflunomide.

## 6. Safety

While generally well tolerated, anti-CD20 mAb therapies have some safety considerations that must be observed. Infusion reactions occur commonly with ocrelizumab therapy, generally within the initial 24 h following administration due to type 2 hypersensitivity reactions and accompanied cytokine release [46,58]. Ocrelizumab therapy is contraindicated in active hepatitis B and C patients, so screening should be considered prior to treatment [11]. The most clinically important adverse events associated with ocrelizumab are infections secondary to immunosuppression. Post-marketing surveillance identified that as many as 30% of patients receiving ocrelizumab show hypogammaglobulinemia, significantly increasing infection risk [59]. The most frequent infections observed in patients treated with ocrelizumab were nasopharyngitis, upper respiratory tract, herpes zoster exacerbations, and urinary tract infections [45,46,47]. Additionally, patients should be examined for vaccination status and refrain from live vaccines during ocrelizumab treatment. It is worthy of mention that there is a lack of data in animal or humans on the teratogenicity of ocrelizumab. Exposure to B-cell depleting agents such as ocrelizumab while pregnant was associated with B-cell depletion in newborns, renal and testicular toxicity, lymphoid follicle formation in the bone marrow, and death in primate models [43]. As a result, contraception is recommended for women in the 6 months after the last infusion. There has also been a tendency to develop neoplasms, specifically different types of breast cancer, presented more frequently in patients receiving ocrelizumab, but a mechanistic role of the antibody exposure has not yet been identified [41]. An important safety concern regarding many MS medications is the risk of progressive multifocal leukoencephalopathy (PML), which has been shown to be attributable to a number of therapies used in treatment of the disease. Although there were no cases of PML reported during ocrelizumab treatment in the clinical trials leading to its approval, a small number of cases have since been reported in post-market monitoring. Many of these patients have been confounded by prior treatment with other MS agents such as standard disease modifying therapies (DMTs) and natalizumab, making a causative link difficult to identify, and the risk/benefit of treatment remains unchanged [60].

Studies investigating the safety of ofatumumab reported adverse events including infection, neutropenia, infusion-related reaction, anemia, thrombocytopenia, cough, and pneumonia [61,62]. In the MIRROR study, AEs were reported to be mild to moderate in severity, and no fatal reactions were reported. Notably, serious AEs were mostly infusion reactions with one case of cytokine-release syndrome in the hours following the first ofatumumab dose. Other less common AEs occurring in single patients were cholelithiasis, hypokalemia, angioedema, and urticaria [56]. In two phase 2 trials of RRMS, no cases of opportunistic infection or incident neoplasm were reported and also no immunogenic responses were observed in any of the patients who received the medication. After FDA approval for ofatumumab in 2014 for treatment of chronic lymphocytic leukaemia (CLL), it has been associated with hepatitis B and PML in post-market monitoring. There has been one reported case of a PML-related death in a patient receiving ofatumumab who had a history of CLL and other comorbid malignant disease [63].

While the anti-CD20 therapies appear to have strong efficacy and favorable safety profiles, there is some concern regarding adverse events when transferring to, or from, another therapeutic such as teriflunomide or interferon beta. Due to the relative infancy of these therapeutics in the treatment of MS, the risk profile of changing to, or from, these medications is currently unknown [64]. However, in head-to-head comparisons of other therapies and the anti-CD20 agents, generally, the efficacy and safety of the agents teriflunomide, interferon beta, or fingolimod, is typically lower, making switching of medications likely to be of benefit overall [64]. Until the antio-CD20 agents have been in widespread use, close monitoring of patients in the months after medication switching is recommended to identify any adverse events.

In conclusion, the most common adverse events reported with anti-CD20 monoclonal antibodies are infusion reactions, which are believed to be due to excessive CDC [65]. When these agents are intravenously administrated, it is essential to have established safety protocols such as premedication with an antihistamine, such as diphenhydramine, and antipyretics such as acetaminophen. To reduce the risk of serious AEs, preventative dosing with glucocorticoids may also be considered.

## 7. Conclusions

Anti-CD20 antibody-mediated, B-cell depleting therapies established a new and promising avenue in the treatment of RRMS and PPMS. They have yet to be evaluated in other forms of MS, such as SPMS, and so their use cannot be supported in this. They are perceived as a promising alternative or complement to the current panel of DMTs. The recorded data delivered worldwide from clinical trials confirms the benefits of these agents on both clinical and imaging assessment of inflammatory activity for patients with RRMS, as well as also confirming benefits in patients with PPMS in reducing disability accumulation [66]. Three major anti-CD20 mAbs have been tested in MS; rituximab, ocrelizumab, and ofatumumab. Broadly, in clinical trials they have been shown to reduce time to disease progression and radiographic findings. Despite these results, there is still further research required to fully evaluate the full risk–benefit ratio of anti-CD20 therapies, as well as to evaluate their impact with extended periods of use, particularly when compared to more established traditional therapies. Further analysis to understand the mechanisms by which B-cell depletion is so effective in MS will also advance the understanding of pathogenesis in MS, as well as identify further strategies for therapeutic approaches.

## Figures and Tables

**Figure 1 brainsci-10-00758-f001:**
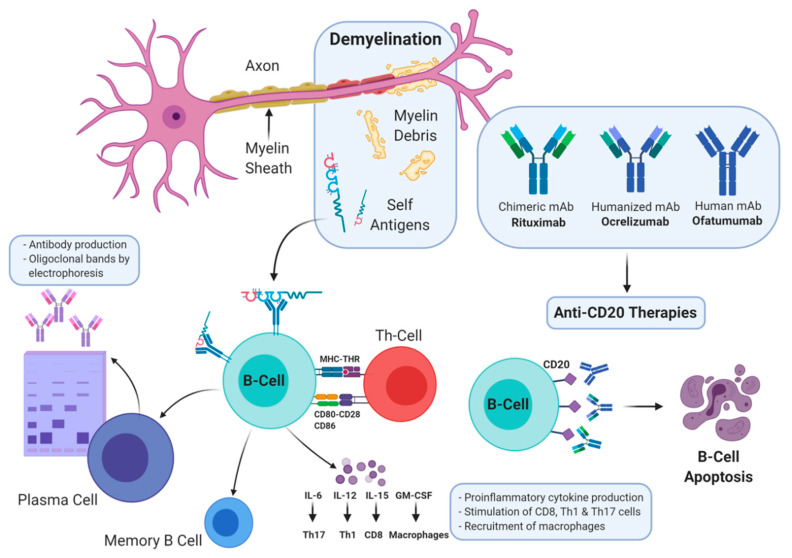
Three Anti-CD20 Antibodies: Rituximab, Ocrelizumab, and Ofatumumab. mAb: monoclonal antibody, Th: T-Helper, IL: Interleukin.

**Table 1 brainsci-10-00758-t001:** A Summary of Clinical Studies Using Rituximab.

Author	Participants	Trial Design	Clinical Findings	MRI Findings	Adverse Effects
Hauser et al., 2008 [33]	104 participants with RRMS	48 wk phase 2, randomized, parallel, double-blind, placebo-controlled study	Decreased number of patients with relapse (24 and 48 wks) and decreased ARR (24 wks only)	Decreased total and new GAD-enhancing lesions with rituximab	Infusion-associated AEs common in Rituximab-treated patients (78.3%). Infection rate similar between groups
Hawker et al., 2009 [34]	439 patients with PPMS	96 wk, phase 2/3, randomized, double-blind, placebo-controlled study	No change in time to disease progression, or the MSFC (with exception of the 25-foot walk)	Decrease T2 lesion volume	8.6% of rituximab patients experienced severe or disabling AEs. 67% of rituximab patients experienced infusion- associated symptoms.
De Flon et al., 2016 [35]	75 patients with clinically stable RRMS	Open Label, uncontrolled phase 2 study	Nil	Decreased number of GAD-enhancing lesions per patient, decreased new or enlarged T2 lesions	Not fully reported. 8% of patients reported severe AEs.
Salzer et al., 2016 [36]	822 patients with MS; 557 RRMS, 198 SPMS, 67 PPMS	Retrospective uncontrolled observational study	EDSS unchanged in RRMS, slight decreases in SPMS and PPMS	Lower disease activity suggested on rituximab	AEs likely underreported; however, 10.8% of patients reported to experience grade 2 or greater AEs.
Zecca et al., 2019 [37]	355 patients with MS	Retrospective uncontrolled observational study	ARR decreased compared to year before rituximab treatment in RRMS and SPMS	Descriptive only	34.5% of patients treated with rituximab had infection-related AEs, 22.7% had non-infectious AEs.
Naeglin et al., 2019 [38]	113 patients with SPMS	Retrospective cohort study	Lower EDSS scores in rituximab cohort	Nil	9% of patients experienced serious AEs
Linden et al., 2019 [39]	272 patients with MS of all subtypes	Retrospective Cohort Study with 43 months mean follow-up	No change in EDSS	T2 lesions correlated with vitamin D levels with rituximab.	38.7% of patients experienced non-infusion AEs, with 7.1% experiencing severe AEs (mostly infectious).

MS: Multiple sclerosis, MRI: Magnetic resonance imaging, GAD: Gadolinium, AE: Adverse event, RRMS: Relapsing remitting MS, PPMS: Primary progressive MS, SPMS: Secondary progressive MS, ARR: Annualize relapse rate, EDSS: Expanded disability status scale, MSFC: MS functional composite.

**Table 2 brainsci-10-00758-t002:** A Summary of Clinical Trials Using Ocrelizumab.

Ocrelizumab					
Author	Participants	Trial Design	Clinical Findings	MRI Findings	Adverse Effects
Kappos et al., 2011 [45]	220 participants with RRMS	48 wk, phase 2, randomized, parallel, double-blind, placebo-controlled study	Lower ARR with ocrelizumab. Reduction in disease activity in placebo and control when crossed over to ocrelizumab.	Decreased GAD-enhancing lesions with 600 & 2000 mg ocrelizumab.	Infusion-associated AEs more common in ocrelizumab. Serious AEs were also reported.
Hauser et al., 2017 [46]	Two studies of 821 and 835 patients with RRMS.	96 wk, phase 3, randomized, double-blind, active-controlled, parallel group studies	ARR reduced in ocrelizumab compared to IFN-β1a	Decrease in new or newly expanding T2 lesions and GAD-enhancing lesions with ocrelizumab	Mild-to-moderate infusion-associated AEs were reported, equally across ocrelizumab and IFN-β1a.
Montalban et al., 2017 [47]	732 patients with RRMS	120 wk, phase 3, double-blind, randomized, placebo-controlled, parallel group study	Reduction in disability at 12 and 24 weeks with ocrelizumab	34% reduction in brain T2 lesions and percentage loss of brain volume decreased with ocrelizumab	Infusion-associated AEs, upper respiratory infections and neoplasms

MS, Multiple Sclerosis; RRMS, Relapsing Remitting Multiple Sclerosis; wk, week; ARR, Annualized Relapse Rate; GAD, gadolinium.

**Table 3 brainsci-10-00758-t003:** A Summary of Clinical Trials Using Ofatumumab.

Ofatumumab					
Author	Participants	Trial Design	Clinical Findings	MRI Findings	Adverse Effects
Sorensen et al., 2014 [55]	38 patients with RRMS	48 wk, phase 2, double-blind, randomized, placebo-controlled study	Decreasing relapse incidence versus placebo (19% vs. 25%)	Decreased new and expanding MRI lesions & total number of T1 GAD-enhancing lesions	Mostly mild-to-moderate-severity AEs
Bar-Or et al., 2014 [56]	231 patients with RRMS	48 wk, phase 2, double-blind, randomized, placebo-controlled study	Nil	Dose dependent decrease in GAD-enhancing lesions with ofatumumab	Increased number of infection-associated AEs in ofatumumab
Hauser et al., 2019 [57]	900 patients with RRMS	30 month maximum parallel Phase 3, double-blind, randomized, placebo-controlled studies (ASCLEPIOS I & II)	Improved relapse rate, and confirmed disease improvement compared to teriflunomide	Decreased number of T1 and T2 gadolinium-enhancing lesions	Equivalent AEs compared to teriflunomide

MS, Multiple Sclerosis; RRMS, Relapsing Remitting Multiple Sclerosis; wk, week; ARR, Annualized Relapse Rate; GAD, gadolinium.

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
