# Peer review of "Anti-CD20 Agents for Multiple Sclerosis: Spotlight on Ocrelizumab and Ofatumumab"

_brainsci, 2020, doi:10.3390/brainsci10100758_

Round 1

Reviewer 1 Report

I would like to thank the authors for their thorough corrections and improvements on their manuscript. Their manuscript is now substantially improved.

Still, I detect serious points of bias and "oversimplification" especially in points where the important aspect of treatment choice and safety is discussed.

More specifically:

1. lines 58-59: "The progression of disability, worsening of disease and radiographical changes in the CNS...": the phrase "worsening of disease" is oversimpification. Please correct appropriate (e.g. inflammatory disease activity or relapse rate). Here it also misses the recently established and important term of NEDA (no-evidence of disease activity), please correct accordingly.

2. line 62: the long-term effects of available drugs are certainly not "limited" as the authors imply (e.g. check the data from alemtuzumab)! Still, the need for new drugs rely on the safety issues, compliance and patient's immune-changes that may render specific drugs -after a period of time- less effective. Please correct.

3. Paragraph in lines 369-379: The authors probably try to bring in the important safety issue of PML (progressive multifocal leukoencephalopathy). Yet their formulation is oversimplified and biased. More specifically: 1) line 372, the phrase "...little data on the if there is any risk when changing to or from these medications" is a dangerous statement! The authors obviously neglect the fact that fingolimod, dimethylfumarate and even ocrelizumab have proven responsible for PML cases, and this after an "euphoric" initial safety profile. 2) Line 373 and 374: the biased expression of "older" medications (referring to teriflunomide and fingolimod) should be removed; old medications can be considered the interferons and glatiramate acetate, yet with favorable safety profile. 3) Line 375: "...lower, making switching of medications likely to be of benefit overall.", the phrase contains bias for changing to anti-CD20, please rephrase. 4) Lines 375-377: "Some clinicians ... of a new radiographic lesion or clinical relapse", pure and oversimplified expression on a topic that is the major discussion thema in MS. Remove or rephrase cautiously!

Overall the above paragraph should be carefully rewritten to include the risk of PML, and should be placed somewhere before the paragraph of lines 364-368 (conclusion paragraph).

4. The "conclusion" paragraph is still no Conclusion. The authors should move lines 390-394 ("Ofatumumab has.... follow in 2021.") in the end of Section 5.3. and lines 394-397 ("Ocrelizumab has ... in clinical trials.") somewhere in the end of section 4.3. Finally, in line 385 please replace the phase "the most promising alternative" with "a promising complementation" to avoid -again- biased statements in favor of anti-CD20.

5. line 69: please recheck syntax.

Author Response

We thank the reviewer for their thorough feedback. We have addressed their revisions and provided a point by point response. Please see the attachment

Reviewer 2 Report

Thank you for the opportunity to review this revised manuscript. I am grateful to the authors for their hard work in revising the text and tables. This is now an excellent and concisely written paper, which should garner interest. All the very best.

Author Response

We thank the reviewer for their input throughout the peer-review process. We are glad they have found it worthy of publication.

Round 2

Reviewer 1 Report

I would like to thank the authors for their constructive efforts and corrections. I now find the manuscript appropriate for publication.

This manuscript is a resubmission of an earlier submission. The following is a list of the peer review reports and author responses from that submission.

Round 1

Reviewer 1 Report

The authors in the present work (Florou, D, et. al.) review the available up-to-date data for two anti-CD20 monoclonal antibodies for MS, namely ocrelizumab and ofatumumab. These 2 antibodies belong to a broader family of anti-CD20 antibodies current under development, all of which are based on the initial effectiveness and translational success of rituximab.

I consider the current work as putatively suitable for publication but only after corrections. As such, I have few major and several minor comments/recommendations which I believe will improve the quality and clinical significance of the review before its publication in Brain Sciences.

More specifically:

Major:

  • Line 223: Change the paragraph "5.3. Safety" into a separate, stand-alone paragraph, namely "6. Safety", for for all antibodies discussed. Include data on rituximab as well and expand the discussion on the common background safety issues for all 3 agents.
  • Conclusion: it contains repetitions from or parts that belong to previous sections (for example Lines 272-277) and is not appropriately written as a concluding remark. Please reform/rewrite. In addition, it contains biased statements (e.g. Line 268-269) that are not scientifically supported. Please correct appropriately.
  • Section 3. (anti-CD20 agents): please organize the section in 2 paragraphs, one for use of anti-CD20 in non-CNS disorders and one for MS. Explain/comment on the rationale behind the development of different anti-CD20 antibodies, i.e. why are they "necessary". Please add a full extra paragraph for the effects and studies of rituximab on MS, as it is the first effective anti-CD20 medication for MS. Please refer to the current clinical evidence (to name few: Naegelin Y et al JAMA Neurol. 2019 Mar 1;76(3):274-281, Zecca C et al Mult Scler. 2019 Oct 1;1352458519872889 etc).
  • Table 1. Please spit in 2 tables, one for each drug (ocrelizumab and ofatumumab). As rituximab is also comparative addressed in the manuscript please add an additional table with all available clinical trials of rituximab in MS.

Minor:

  • Title: as Rituximab is also a potent -and long available anti-CD20 agent- the title could be more general, for example "Anti-CD20 agents for MS: focus on ocrelizumab and ofatumumab".
  • Abstract, Line 15: the word "disregarded" for B-cells is too strong, please change to "underestimated"
  • Abstract, Lines 17-18: the expression of CD-20 defines specific subpopulations of B-cells (pro-B-cells, immature B-cells, mature B-cells and memory B-cells) and is not "uniquely" expressed as a result of MS-immunopathological processes. Please rephrase to avoid misunderstanding.
  • Abstract (general comment): please refer briefly to the positive effects of Rituximab in MS.
  • Lines 39-40: please add appropriate references for the data given.
  • Lines 55-56: "B cells have been defined as regulatory B cells (secrete IL-10, IL-35) and in MS as pro-inflammatory B cells...": not clear what is meant by "defined"; is it meant "B cells are grouped as either regulatory B cells (secrete IL-10, IL-35) or as pro-inflammatory B cells - as is the case in MS- ..."?
  • Line 58: please correct to "In addition, B cells, next to other antigen-presenting cells, play a highly effective role..."
  • Lines 63-65: Repetitions, please correct accordingly.
  • Line 66: "Depletion of B cells is therefore..." the authors imply a deleterious and neurodestructive effect of B-cells that is not clearly supported by the lines before 66. Please add data that support a direct connection between B-cell activation with IgG secretion to CNS lesions in MS (e.g production of antibodies against myelin components, complement on-site activation after antibody-binding to targets, induction of apoptosis, attraction of macrophages etc).
  • Line 74-75: "...CD19 and CD20 have provided to be viable targets for selective targeting for monoclonal antibodies...": why is this so? Please add a short comment in support of its significance.
  • Line 131: how long is the half-life for ocrelizumab?
  • Lines 143-145: please report the actual percentages of infusion-related adverse effects in the 600 and 2000mg groups.
  • Line 158: "... the progression of disability was lower" please write if this was also statistically significant, as the differences are low. Taken this as an example, please report more accurate any significant or not significant difference in the manuscript, where missing.
  • Line 184: "...self-administered...", presumably it is meant subcutaneously, please correct appropriately. Please also write the administration interval.
  • Line 185-186: "In the face of.... infusion". This -written as such- is a biased statement: please remove, or rephrase.
  • Line 199: "After 24 weeks in two groups exchanged treatment groups." Note clear, please rephrase.
  • Line 209: are the different AEs statistically significant?
  • Line 221: during the revision process, ofatumumab is approved for MS by FDA (for clinically isolated syndrome, relapsing-remitting MS, and active secondary progressive MS), please update appropriately.

Author Response

Q: Nice review of the filed that might require modest editing for English A. We thank the reviewer for their feedback.

A: The manuscript has undergone editing by our English speaking colleagues from Australia.

Reviewer 2 Report

Nice review of the filed that might require modest editing for English

Author Response

Reviewer 1

Q: Nice review of the filed that might require modest editing for English

A: We thank the reviewer for their feedback. The manuscript has undergone editing by our English speaking colleagues from Australia.

Reviewer 3 Report

Florou et al reviewed the state of art of Ocrelizumab and Ofatumumab use for Multiple Sclerosis

The review has some merits but I feel that it needs profound revision

First, the authors did not described well the fundamental role of B cells in the pathogenesis and development of the disease.

Many questions about the role of B cells in MS pathogenesis remain still unanswered. What factors drive B cells into the CNS, through which pathways do they travel, and is this cell traffic persistent or transient? When during disease do B cells populate the CNS, and are there other CNS niches in which B cells thrive? How do immune cell aggregates in the tertiary ectopic follicles contribute to MS pathology?

Then, which are the open questions about the choice of anti-CD20 drugs?

In perspective which is the role of anti CD20 in the therapeutic scenario of MS?

Bibliography need to be updated and other reviews in such field should be cited

Author Response

We thank the reviewer for their insightful feedback - we have integrated your suggestions to the manuscript. Please see the attached document for a point by point response.

Reviewer 4 Report

The subject of this review is timely and the topic of great interest to clinicians and neuroscientists. Since the authors submitted this article there has been a further announcement from the FDA granting approval for Ofatumumab in active SPMS- this further highlight the timeliness of the manuscript.

The authors have taken a rather passive role in describing the published data to date regarding these two new DMTs for MS. There has been no effort to compare the efficacy data to that previously published for other approved therapies for progressive MS and no comment with regards to switching to- or from- ocrelizumab from other DMTs. This second point is a hugely important concern for the MS patient care team. The current version of this manuscript contains many basic errors that urgently need correcting before it is publication ready. Can I direct the authors to a selection of concerns/ corrections...

Line (L) 33: this sentence is muddled. Replace progressive effect with ‘worsening’ and reserve the term ‘progression/ progressive’ for the relevant description of disease stage.

L35: You must know that PRMS is not a recognised clinical term since Lublin’s revisions in 2013! PRMS is not a distinct disease subtype.

You must update your reference list and use the terms progression with inflammatory activity as these are the definitions used in the trial’s papers and FDA/ agency decision makers. You should take this opportunity to describe how patients are currently monitored as an explanation of EDSS and radiographic evidence of central inflammation are central to the testing and prescribing of these drugs.

L39: You must include a reference to support your assertion that 60% of people with MS are wheelchair bound after 20 years. I suspect this figure is from older studies and the demographics are now rapidly shifting since the advent of efficacious treatments (people are diagnosed at an earlier age, for example).

L41: This statement is disingenuous. Treatment for PMS may be limited compared to RMS but siponimod, ocrevus, mitoxantrone, cladribine all approved by FDA as they show efficacy in active progressive disease.

L45: sites of neuronal, myelin and BBB damage and glial activation

L51: this sentence needs rewriting as earlier in the paragraph you state that B cells are a component of the CNS infiltrate (and have been shown since the 1990’s as have auto-antibodies and Ig+ plasma cells, which you do not describe).

L64: citations- none of these papers demonstrate B cells in MS lesions. Other references required here.

L72: A distinction between B-cell lineage selective and B cells plus many other cell-expressed markers (such as CD32, CD35) needs to be made. CD27 is a useful marker of the memory B cell pool when combined with other B cell phenotypic markers.

L80: Please consider the evidence for CD20+ T cells (https://pubmed.ncbi.nlm.nih.gov/30561509/ ) or at least acknowledge their description by some.

L101: please be consistent in your use of abbreviations- is it RRMS or RMS?

L112: Specifically approved for the use in active progressive MS!!

L128: you refer to CD20+ T cells here, so why not earlier (see comment re: L80).

L131: ‘beaks’ !

L154: defining EDSS and other clinical trial end-points such as relapse rate and neuroimaging in your intro would have been helpful.

L272: ‘accretion’ is not the appropriate term. You presumably mean ‘accumulation’.

L273: Please take the opportunity to include the latest update from the FDA in any revisions.

Author Response

(The authors gave the same response as above.)
